# Coating Methods of Carbon Nonwovens with Cross-Linked Hyaluronic Acid and Its Conjugates with BMP Fragments

**DOI:** 10.3390/polym15061551

**Published:** 2023-03-21

**Authors:** Sylwia Magdziarz, Maciej Boguń, Justyna Frączyk

**Affiliations:** 1Institute of Organic Chemistry, Faculty of Chemistry, Lodz University of Technology, Zeromskiego 116, 90-924 Lodz, Poland; 2Łukasiewicz—Lodz Institute of Technology, Sklodowskiej-Curie 19/27, 90-570 Lodz, Poland

**Keywords:** hyaluronic acid, cross-linking, BMP fragments, citric acid, BDDE, carbon nonwoven, coating

## Abstract

The cross-linking of polysaccharides is a universal approach to affect their structure and physical properties. Both physical and chemical methods are used for this purpose. Although chemical cross-linking provides good thermal and mechanical stability for the final products, the compounds used as stabilizers can affect the integrity of the cross-linked substances or have toxic properties that limit the applicability of the final products. These risks might be mitigated by using physically cross-linked gels. In the present study, we attempted to obtain hybrid materials based on carbon nonwovens with a layer of cross-linked hyaluronan and peptides that are fragments of bone morphogenetic proteins (BMPs). A variety of cross-linking procedures and cross-linking agents (1,4-butanediamine, citric acid, and BDDE) were tested to find the most optimal method to coat the hydrophobic carbon nonwovens with a hydrophilic hyaluronic acid (HA) layer. Both the use of hyaluronic acid chemically modified with BMP fragments and a physical modification approach (layer-by-layer method) were proposed. The obtained hybrid materials were tested with the spectrometric (MALDI-TOF MS) and spectroscopic methods (IR and 1H-NMR). It was found that the chemical cross-linking of polysaccharides is an effective method for the deposition of a polar active substance on the surface of a hydrophobic carbon nonwoven fabric and that the final material is highly biocompatible.

## 1. Introduction

Many unique molecules, materials, and even technological solutions, such as the hydrophobic surface of the lotus leaf, the ability of the cactus to store water molecules from mist, and the ability of owl feathers to suppress noise, are observed in nature. The inspiration gained from nature can be seen in many areas of science and industry, including the development of some breakthrough materials and medical devices. The gecko (*Gekko gecko*) has a remarkable ability to climb a variety of natural surfaces. Its feet plastically adapt to the structure of a surface due to a layered system of foot traction. Thanks to the significant research on gecko-like dry adhesive surfaces during the last 20 years, including both mechanical measurements of the adhesive characteristics and the theoretical modeling of the adhesive mechanism, the production of synthetic dry adhesive surfaces has become understood and the possible application of such binders has been tested. A major breakthrough was the creation of an innovative tissue adhesive that mimics the scaly surface of a gecko’s foot by using small nanostructures to which a thin glue is then applied, creating a biodegradable bandage for organ and tissue repair [1]. Scientists have also seen medical potential in the octopus’s tentacles. The octopus (*Octopus vulgaris*) uses its muscular tentacles to expand and contract around its prey in a mechanism resembling an ultra-thin graft suction cup. The University of Illinois developed a suction cup device that is capable of lifting a graft with little or no pressure [2]. The proposed solution uses a temperature-responsive hydrogel layer and is able to eliminate one of the biggest risks in the transplant field—damage to or contamination of the tissue prior to transplantation. Brightly colored insects or animals, such as a butterflies, beetles, and peacocks, can have hues of blue, green and purple. This unique coloring is based on many different factors, including the nanostructure of the scales or feathers, the number of these layers, and the light angle. This information served as an inspiration to produce opal-like photonic crystals that can change color in response to not just light but also temperature, chemical changes, and strain [3]. Such a solution could be applied to a number of medical applications, including biosensing technologies used to detect respiratory viruses, bio-monitoring used to improve or enhance physical performance, or even healthcare safety used to detect surface contamination.

The modern concept of tissue reconstruction is based on the search for solutions to regenerative medicine problems and for naturally occurring materials with unique properties and their direct use or inspiration. One such material is hyaluronic acid, also commonly named hyaluronate. It is a polysaccharide that occurs in all body fluids and tissue types. It is found in large quantities in the vitreous body of the eye or synovium. It is a compound that combines unique properties and a simple chemical structure. It is composed of D-glucuronic acid and N-acetyl-D-glucosamine [4] linked with β (1-3) and β (1-4) glycosidic bonds. The most common division of this biopolymer is based on its molecular weight: low (<20 kDa), medium (100–300 kDa) and high (>1000 kDa). Importantly, the structure of hyaluronic acid is identical regardless of whether it is produced by bacteria, birds or mammals. However, its biological activity is determined by its molecular weight [5]. Low-molecular-weight HA stimulates Toll-like receptors to release pro-inflammatory cytokines [6], chemokines and other molecules responsible for tissue proliferation [7]. Medium-molecular-weight HA induces β-defensin 2 receptors of the skin epithelium, displaying the ability of tissue regeneration [8,9], and also stimulates the production of nerve cell growth factors [10]. Finally, high-molecular-weight HA shows anti-inflammatory activity and immunosuppressive properties [11]. Hence, HA-based biomedical engineering solutions consider the optimal molecular weight with respect to the effectiveness of the therapy in addition to patient safety [12]. Hyaluronic acid is not only a component of the extracellular matrix (ECM) but also surrounds the cells in the pericellular matrix (PCM) [13]. In addition, it is a bioinert material, interacting with ECM proteins that have important biological functions [14]. Hence, the application of this biopolymer in medicine has been extensive. 

The cosmetics industry is the leader in producing ready-made products containing hyaluronic acid. In 2019 alone, this market was worth as much as 8 billion USD. For the past few decades, HA has been mainly used as an ingredient in moisturizing cosmetics, and data on HA’s diverse biological effects and gravity-dependent effects have supported its use in newer applications. Low-molecular-weight hyaluronic acid has been shown to have transdermal properties [15] because it retains water in the deeper layers of the skin, which results in the shallowing of wrinkles and better skin tone. In contrast, HA with a molecular weight of >300 kDa does not penetrate deep into the skin; rather, it remains on the surface, encapsulating water molecules from the air and creating an invisible barrier [16]. The positive influence of hyaluronic acid on skin regeneration and renewal is explained in two ways. First, N-acetyl-D-glucosamine (one of the HA components) stimulates the production of new HA [17]. Secondly, the CD44 receptor induces HA biosynthesis in response to a reduced concentration of HA in the skin [18]. HA is used in the treatment of psoriasis and other skin diseases. Conventional methods of treating such disorders still have some limitations, mainly related to inadequate skin penetration or the induction of off-target responses. Therefore, hyaluronic acid has begun to be used in the form of nanoparticles or hydrogels as alternative treatments. In vivo studies on hyaluronic acid nanoparticles (HA NPs) have demonstrated the high efficacy and long-lasting and specific action of this form of HA [19,20]. Hyaluronic acid-based hydrogels have been tested both as carriers for a standard drug against psoriasis (methotrexate (MTX) [21]) and in the form of hydrogels with different molecular weights [22]. These experiments confirmed the effectiveness of targeted therapy and improved drug absorption. The potential of hyaluronic acid-based hydrogels was also tested for wound healing. Such materials, enriched with fibroblast growth factor (βFGF) [23] or active substances (gentamicin [24], curcumin [25]), have demonstrated positive results with regard to the quality of reconstructed skin (reduced scarring) and reduced recovery time, including after burns. The applicability of hyaluronic acid has also been tested in the form of nanocarriers. HA nanoparticles, designed for targeted therapies in cancer treatment, have been obtained with both the bottom–up and top–down approaches. The first group includes nanocarriers based on methacrylated hyaluronic acid (MAHA) [26]. The second approach uses nanocarriers based on graphene oxide (GO) [27] or perfluorinated compounds [28] decorated with hyaluronic acid. In all cases, the delivery system has been found to work with the expected efficiency, enabling doxorubicin (DOX) to be delivered. DOX release was found to occur under controlled conditions, and the duration of action was extended up to 12 h. The cytotoxicity of such carriers against cancer cells increased almost threefold without inducing a toxic systemic response.

Hyaluronic acid has found a particularly wide range of applications in tissue engineering. The wide range of application forms, i.e., fibers, hydrogels, nanoparticles, and injections, provides the opportunity to test HA-based materials in regeneration of various tissues. The injectable form has proven successful, for example, in the treatment of dysphonia and osteoarthritis. In the first case, the injection solution was based on hyaluronic acid oxide and a modified polyurethane emulsion. This kind of material not only increases the safety and the precision of the application process but also affects the proliferation of new cells and therefore promotes the scarless healing of vocal folds [29]. For osteoarthritis therapy, viscosupplementation with exogenous hyaluronic acid has been found to have a direct effect on accelerating chondrocyte proliferation, reducing cartilage degradation and reducing the possibility of local inflammation [30,31]. As a result, such a material not only has the therapeutic effect of improving the lubrication of damaged joints but also reduces pain and delays the need for prosthetic surgery. Another interesting approach to the use of hyaluronic acid in tissue engineering is the creation of active scaffolds for tissue regeneration. As one of the components of ECM, hyaluronic acid has been used in its “raw” form to create nanofibers, which are the base of scaffolds. The electrospinning method used in this case allows for the manipulation of the process parameters and, consequently, the parameters of the final product. The resulting material, characterized by high porosity and swelling abilities, also shows mechanical properties with values similar to natural soft tissues [32]. A bioink based on hydrazone-cross-linked hyaluronic acid and photo-cross-linked gelatin methacrylate was used to obtain scaffolds with a 3D printing method. The resulting hydrogel was characterized by a double network, which provided increased mechanical strength and also promoted the adhesion and proliferation of bone marrow stem cells [33]. In addition to the therapeutic effect of accelerating the proliferation of new cells and tissue reconstruction, the possibility of creating an HA-based tissue adhesive has also been investigated. Hydrogels with varying degrees of catechol [34] or dopamine [35] substitution have been obtained. The result are materials that exhibit both excellent tissue adhesion and a controlled degradation profile. 

These medical applications of hyaluronan are linked not only by the material itself but also by the cross-linking process. This is caused by the formation of a chemically ordered network of hyaluronic acid, which increases its molecular weight, increases its residence time in the body without degradation, and alters its viscoelastic properties with respect to un-cross-linked products [36,37]. Most often, the process of hyaluronic acid cross-linking is performed by adding a cross-linking agent to HA in solution to obtain a solid gel, which is finally dried [38]. This leads to a product whose properties depend on the degree of cross-linking and particle size. From a medical perspective, cross-linking is believed to improve the effectiveness of formulations for specific applications (viscosupplementation [39], skin augmentation, and so on [40,41]). In addition, it is also possible to increase biocompatibility, regulate adhesion, or improve biological responses by modifying hyaluronan with additional substances or creating functional layers over the carriers [42,43]. The homogeneous coating of materials with hyaluronic acid increases the application potential of known implants by giving them additional properties, such as antibacterial and osteoinductive properties [44]. 

One of the latest trends in regenerative medicine is obtaining hybrid materials based on fibrous matrices. Those materials are characterized by high mechanical strength, required porosity, biocompatibility, and the ability to inhibit bacterial growth. Among many fibrous materials, carbon nonwovens could be considered a special case. They have excellent conductivity, high chemical resistance, and a very good mechanical strength-to-weight ratio. Additionally, they could be the basis of a biomaterial with osteoconductive and osteoproductive properties. It has been already proven that carbon fibers (CFs) are very good reinforcing materials and could work under different conditions. These features could also be applied in the medicinal field. There are known examples of biomaterials based on composites of CFs and poly-ether-ether-ketone (PEEK) useful in orthopedics [45,46,47]. Another application field of CFs is in heart valves [48] or biomaterials for cardiac tissue regeneration [49]. Neurology is also a promising field. Carbon fiber-based electrodes have been used to detect the changes in the neuronal activity of brain tissue [50], as well as in implantable depth neural electrodes [51]. The biocompatibility of the fibers is the common feature of the above-mentioned applications [52,53], and the biocompatibility is determined by the type of precursor used for the fabrication of the fibers [54].

With these factors under consideration, we attempted to develop hybrid materials based on carbon nonwoven fabrics coated with cross-linked hyaluronic acid and to apply an optimal cross-linking method to obtain carbon nonwovens coated with a conjugate of cross-linked hyaluronic acid with BMP fragments. It was expected that the coating of carbon nonwovens with cross-linked HA would improve the biocompatibility of the carbon materials and would also better condition their biological activity. As a part of the research planned, it was necessary to select the optimal procedure for cross-linking hyaluronic acid from the point of view of its applicability for coating hydrophobic carbon nonwovens. An additional modification with BMP fragments ensured that the final hybrid materials would be characterized by the increased adhesion and proliferation of osteoblasts and/or chondrocytes. We also tested the possibility of using the formation of HA conjugates with BMP fragments based on the formation of permanent chemical bonds (chemical method) and using conjugates formed through a network of weak bonds (physical method). The efficiency of the conducted functionalizations was confirmed with spectroscopic (IR and ^1^H NMR) and mass spectrometric (MALDI-TOF MS) methods.

## 2. Materials and Methods

Most of the used reagents (1,4-diaminobutane, N-(3-fimethylaminopropyl)-N′-ethylcarbodiimide hydrochloride (EDC), N-hydroxysuccinimide (NHS), citric acid (CA), 1,4-butanediol diglycidyl ether (BDDE), α-cyano-4-hydroxycinnamic acid (HCCA), and Fmoc-protected amino acids) were purchased from Sigma-Aldrich Poznan, Poland. Hyaluronic acid (HA) was obtained from Contipro a.s., Dolni Dobruc, Czech Republic (product name: HyActive; molecular weight: 80–130 kDa). NaOH and HCl were obtained from Eurochem BGD Sp. z o.o., Tarnow, Poland. N,N-Diisopropylethylamine (DIPEA), N-methylmorpholine (NMM), and piperidine were purchased from Carl Roth GmbH + Co. KG, Karlsruhe, Switzerland. Trifluoracetic acid (TFA), needed for the preparation of TA30 (0.1% TFA), was obtained from Fluorochem Ltd., Hadfield, United Kingdom. The reagents used during peptide synthesis, Fmoc-protected amino acids and 2-chlorotrityl resin, were purchased from Sigma-Aldrich Poznan, Poland. Dimethylformamide (DMF) was obtained from Avantor Performance Materials Poland S.A., Gliwice, Poland. 4-(4,6-Dimethoxy-1,3,5-triazin-2-yl)-4-methylmorpholinium-4-toluene sulfonate (DMT/NMM/TosO-) was developed at and obtained from the Institute of Organic Chemistry, Technical University of Lodz, Poland. 

All prepared samples of the carbon nonwovens had dimensions of 2 × 2 cm. 

### 2.1. Cross-Linking Process Choosing an Optimal Procedure

#### 2.1.1. Cross-Linking of HA with 1,4-diaminobutane

The cross-linking reaction between HA and 1,4-diaminobutane required the activation of the carboxylic groups of HA, which was carried out by a reaction with the carbodiimide EDC in the presence of NHS. In this procedure, HA and NHS were dissolved in distilled water (sustaining an NHS/COOH molar ratio of 4:1). Then, the pH was adjusted to 5.4 via the addition of a 2 M NaOH solution. After 15 min of stirring, the EDC and NHS were slowly added until the EDC/NHS/COOH molar ratio was 10:4:1. The mixture was stirred at room temperature for 4 h. Once a homogeneous solution was obtained, 1,4-diaminobutane was added with vigorous stirring at a 1:1 (HA:1,4-diaminobutane) molar ratio. The mixture was stirred (450–550 rpm) for 1 h at room temperature. 

#### 2.1.2. Cross-Linking of HA with Citric Acid Procedure I

HA (0.1 g) was dispersed in water (20 mL). In this polymeric solution, the cross-linking agent citric acid (CA) (0.0252 g) was dissolved via stirring at room temperature for 20 min. The molar ratio between the primary –OH groups of HA and CA was 10:1. The sample was then lyophilized.

#### 2.1.3. Cross-Linking of HA with Citric Acid Procedure II

Samples were prepared by dissolving HA in distilled water at 40 °C in two flasks in order to obtain 1% concentration solutions. Then, 5% and 20% (*w*/*w*) concentration solutions with citric acid were added into the flasks and continuously stirred. The reaction was conducted at room temperature. For both cross-linking agent concentrations, we prepared 2 samples each drying method. Both combinations of HA and CA concentrations (1% HA with 5% CA and 1% HA with 20% CA) were either left for two days at room temperature to dry or lyophilized in order to observe differences between them.

#### 2.1.4. Cross-Linking of HA with BDDE

HA (0.1048 g) was dissolved in 1% NaOH (9.25 mL) via stirring at room temperature until the mixture was homogeneous. Then, the 1,4-butanediol diglycidyl ether (BDDE) cross-linking agent (0.2170 g) was added. It was mixed for 4 h at 45 °C. The sample was then diluted with distilled water (10 mL) and evaporated using a rotary vacuum evaporator. Then, the pH of the obtained sample was adjusted (up to 7.4) by adding PBS (phosphate-buffered saline). The sample was lyophilized.

### 2.2. Cross-Linking of HA on the Carbon Nonwoven Fabric

In order to obtain the most optimal results (rigid layer of cross-linked HA on the surface of the carbon material), a set of samples was prepared by manipulating the concentrations of the hyaluronic acid and cross-linking agent.

#### 2.2.1. Cross-Linking of HA on the Carbon Nonwoven Fabric Procedure with BDDE

HA (0.1 g) was dissolved in a 1% solution of NaOH (9.25 mL) via stirring at room temperature until a homogeneous solution was obtained. Then, the wetted carbon material (treatment with DMF, DMF:H2O (1:1), and H2O; 15 min each [55]) was placed in a flask with the HA solution for 15–20 min. Subsequently, BDDE (0.217 g) was added and stirred for another 4 h in a water bath at 45 °C. Next, the solution was diluted by adding 10 mL of distilled water, and the PBS solution (pH 7.4) was added. The sample was lyophilized. To test the effect of HA concentration, a set of BDDE-cross-linked samples was prepared using HA at different concentrations: 0.5%, 1% and 2%.

#### 2.2.2. Cross-Linking of HA on the Carbon Nonwoven Fabric Procedure with CA

The first experiments were undertaken for concentrations of hyaluronic acid about 10× lower than the concentration of the cross-linking agent, i.e., 0.5% for 5% CA and 2% for 20% CA. A portion of hyaluronic acid was dissolved in distilled water in a water bath at 40 °C. Then, the wetted carbon material (according to the standard procedure of DMF, DMF:H_2_O (1:1), and H_2_O; 15 min each) was immersed in a flask with HA and stirred for 1 h. After this time, 20 mL of 5% and 20% aqueous citric acid solutions was prepared and added to the flasks. Stirring was continued for another hour. After this time, the samples were dried in two ways: (1) freeze-drying and (2) in a laminar air flow chamber and desiccator.

### 2.3. Synthesis of the Polysaccharide–Peptide Conjugate

#### 2.3.1. Synthesis of BMP2 (241-250) Peptide Fragment H-KRMVRISRSL-OH

In the first step of the synthesis, Fmoc-Leu-OH (0.530 g and 1.5 mmol) was attached to 0.5 g of 2-chlorotrityl resin according to the procedure described in the Appendix A. N-N-Diisopropylethylamine (DIPEA) (522 µL and 3 mmol) was used in the reaction. The degree of resin loading was determined according to the general procedure described in the Appendix A. The calculated resin loading was 0.4 mmol/g. Next, the deprotection of the Fmoc group was carried out on the resin containing the C-terminal amino acid according to the general procedure described in the Appendix A. For the synthesis of the final peptide, the following were used: Fmoc-Ser(tBu)-OH (0.460 g and 1.2 mmol), Fmoc-Arg(Pbf)-OH (0.209 g and 1.2 mmol), Fmoc-Ile-OH (0.424 g and 1.2 mmol), Fmoc-Val-OH (0.407 g and 1.2 mmol), Fmoc-Met-OH (0.446 g and 1.2 mmol) and Fmoc-Lys(Boc)-OH (0.562 g and 1.2 mmol) according to the peptide sequence. 4-(4,6-Dimethoxy-1,3,5-triazin-2-yl)-4-methylmorpholinum toluene-4-sulfonate (DMT/NMM/TosO-) (0.496 g and 1.2 mmol) and N-methylmorpholine (NMM) (660 µL and 3.6 mmol) were used for amino acid addition. Reactions were carried out according to the general procedure described in the Appendix A. The removal of the Fmoc protecting group was carried out according to the general procedure described in the Appendix A using a 25% piperidine solution. After the completion of the reaction, the final product was cleaved according to the general procedure described in the Appendix A. The final product was obtained with >99% purity. Its structure was confirmed with MS, *m*/*z* = 313.7092 and *m*/*z* = 313.7027, corresponding to [M+4H]^4+^ of the expected product with M = 1251.50 g/mol.

#### 2.3.2. Synthesis of BMP9 (361-370) Peptide Fragment H-FFPLADDVTP-OH

In the first step of the synthesis, Fmoc-Pro-OH (0.506 g and 1.5 mmol) was attached to 0.5 g of 2-chlorotrityl resin according to the procedure described in the Appendix A. DIPEA (522 µL and 3 mmol) was used in the reaction. The degree of resin loading was determined according to the general procedure described in the Appendix A. The calculated resin loading was 0.4 mmol/g. Next, the deprotection of the Fmoc group was carried out on the resin containing the C-terminal amino acid according to the general procedure described in the Appendix A. For the synthesis of the final peptide, the following were used: Fmoc-Thr(tBu)-OH (0.477 g and 1.2 mmol), Fmoc-Val-OH (0.407 g and 1.2 mmol), Fmoc-Asp(OtBu)-OH (0.494 g and 1.2 mmol), Fmoc-Ala-OH (0.374 g and 1.2 mmol), Fmoc-Leu-OH (0.424 g and 1.2 mmol), Fmoc-Pro-OH (0.405 g and 1.2 mmol), and Fmoc-Phe-OH (0.465 g and 1.2 mmol) according to the peptide sequence. DMT/NMM/TosO- (0.496 g and 1.2 mmol) and NMM (660 µL and 3.6 mmol) were used for amino acid addition. Reactions were carried out according to the general procedure described in the Appendix A. The removal of the Fmoc protecting group was carried out according to the general procedure described in the Appendix A using a 25% piperidine solution. After the completion of the reaction, the final product was cleaved according to the general procedure described in the Appendix A. The final product was obtained with a purity of 90%. Its structure was confirmed with MS, *m*/*z* = 374.5092 and *m*/*z* = 374.5142, corresponding to [M+3H]^3+^ of the expected product with M = 1120.54 g/mol.

#### 2.3.3. Chemical Modification of HA with BMP2 (241-250) Peptide Fragment H-KRMVRISRSL-OH by Using EDC and NHS

HA (1.08 g) was dissolved in 45 mL of water to dissolve/swell the polysaccharide. The solution was continuously stirred for 45 min. Then, 45 mL of a 1 mg/mL aqueous peptide solution was added to the flask, along with EDC (3 mmol and 0.4658 g) and NHS (3 mmol and 0.3453 g). The pH of the reaction mixture was set to 5.5 (adjusted with 1 M of HCl or 1 M of NaOH). After 45 min of stirring, the pH of the mixture was checked again and adjusted to a value of about 6–7. The conjugate synthesis reaction was carried out for another 5 days. After 5 days of reaction, the reaction product was precipitated with rectified ethanol in an ice bath. Next, 270 mL of ethanol was dropped into the flask. The obtained solution was centrifuged (4000 rpm for 10 min at 4 °C), and the formed precipitate was successively washed with a 1:4 mixture of H_2_O:EtOH and absolute ethanol. The solid conjugate precipitate was lyophilized.

#### 2.3.4. Chemical Modification of HA with BMP9 (361-370) Peptide Fragment H-FFPLADDVTP-OH by Using Triazine Based Coupling Reagent

HA (1.08 g) was dissolved in 45 mL of water to dissolve/swell the polysaccharide. The solution was continuously stirred for 45 min. Then, 45 mL of a 1 mg/mL aqueous peptide solution was added to the flask, along with DMT/NMM/TsO^-^ (3 mmol and 1.239 g) and NMM (3 mmol and 330 µL). The pH of the reaction mixture was set to 6.5 (adjusted with 1 M HCl or 1 M NaOH). The conjugate synthesis reaction was carried out for another 2 days. After 2 days of reaction, the reaction product was precipitated with rectified ethanol in an ice bath. Then, 270 mL of ethanol was dropped into the flask. The obtained solution was centrifuged (4000 rpm for 10 min at 4 °C), and the formed precipitate was successively washed with a 1:4 mixture of H2O:EtOH and absolute ethanol. The solid conjugate precipitate was lyophilized.

### 2.4. Synthesis of the Hybrid Material—Carbon Nonwoven Fabric Coated by the Conjugates of HA and BMP Fragments

#### 2.4.1. Layer-by-Layer Method—Physical Binding of the Peptide

First, HA was dissolved in water in order to obtain its required concentration (in the range of 0.5–2%). Then, wetted carbon nonwoven was put into the flask, and an H-FFPLADDVTP-OH solution at a concentration of 1 mg/mL (10 mL) in water was added. The cross-linking process followed the procedures described before depending on the used cross-linking agent.

#### 2.4.2. Using Polysaccharide–Peptide Conjugate Chemical Bond between PEPTIDE and HA

In order to obtain a hybrid material with chemical binding between the peptide and HA, the procedure verified for native hyaluronan was adopted. In this case, modified hyaluronan (described in Section 2.3.3) was used. In the case of cross-linking (by both citric acid and BDDE), the wetted carbon material was placed into a flask with a homogenous solution of HA at the required concentration before the cross-linker was added. 

### 2.5. Analytical Techniques

#### 2.5.1. MALDI-TOF MS

The matrix used in the analysis was HCCA (α-cyano-4-hydroxy cinnamic acid) (Sigma-Aldrich, Poznan, Poland) in amount of approx. 10 mg, which had to be dissolved in 500 µL of TA30 (0.1% trifluoroacetic acid (TFA) aqueous solution). Small amounts of the test samples were dissolved in 100 µL of 0.1% TFA. Then, 2 µL of the matrix solution was added to 2 µL of each tested sample solution. Pre-mixed equal volumes (1 µL each) of the sample and matrix solutions were applied onto a ground steel target. The crystallization time of the matrix with the sample was approximately 1 h. Mass spectra were then obtained in the range of 1000–3500 *m*/*z* for each of the samples. Measurements were conducted with a Bruker MALDI–MS ultrafleXtreme spectrometer (Bruker, Karlsruhe, Germany).

#### 2.5.2. ^1^H NMR

Around 10 mg of each sample was taken and added to 1 mL of deuterated water (D_2_O). For all samples, the same scanning conditions were applied: 16 scans/sample. Measurements were conducted with a Bruker Avance II Plus 700 MHz (Bruker, Karlsruhe, Germany). All NMR spectra were analyzed and interpreted using the MestReNova 9.0 software.

#### 2.5.3. FT-IR

A small amount of solid substance was taken for measurement (samples in solution are not adequate for this kind of measurement because water peaks would be significant and cover other significant ones). For the solid samples, more than one measurement was performed in order to check the homogeneity of the coated samples. As a baseline, we used a measuring table. For all samples, the same scanning conditions were applied: 26 scans/sample. All measurements were conducted with a Bruker FT-IR ATR Alpha spectrometer (Bruker, Karlsruhe, Germany).

## 3. Results and Discussion

### 3.1. Choosing the Most Effective HA Cross-Linking Method

In order to select the most optimal method for cross-linking hyaluronic acid on the target fibrous materials, cross-linking tests were carried out using different cross-linking agents with different concentrations. The confirmation of obtaining the required structures was carried out on the basis of structural analyses using the following spectroscopic methods: ^1^H NMR and FT-IR.

#### 3.1.1. ^1^H NMR Spectra Analysis

According to the literature data [56], the characteristic signals in the ^1^H NMR spectrum for unmodified hyaluronic acid are between 3 and 4 ppm, specific to protons in the sugar ring. A signal between 4.3 and 4.5 ppm, corresponding to two anomeric protons, is also characteristic. The protons of the N-acetyl group of hyaluronate correspond to a signal between 1.8 and 2.0 ppm. The signal integration for pure HA is 2.0 (anomeric protons): 10.0 (sugar ring protons): 3.0 (N-acetyl group protons). Figure 1 shows the ^1^H NMR spectrum obtained for hyaluronic acid used for further modifications.

All presented NMR spectra were analyzed and interpreted using the MestReNova 9.0 software. The obtained spectrum matched the literature data. Signals from protons in the sugar ring could be observed at 3.46 and 3.65 ppm, and the characteristic signal of the N-acetyl group of hyaluronan showed a shift of 1.96 ppm.

##### Cross-Linking with 1,4-diaminobutane

In the first step, 1,4-diaminobutane was used as the cross-linking agent. Figure 2 shows the ^1^H-NMR spectra of hyaluronic acid (substrate) and the product of cross-linking HA with 1,4-diaminobutane. The product spectrum showed characteristic peaks in the 2.0 to 2.70 ppm range. A strong signal at 2.07 ppm indicates the attachment of an amine to HA. The peak at 2.59 ppm is attributed to an amine/amide group formed by cross-linking or present in the cross-linking agent used [57]. The strong signal at around 5.0 ppm came from the used solvent (deuterated water).

##### Cross-Linking with Citric Acid

Another tested cross-linking agent was citric acid. Figure 3 shows a comparison of the ^1^H-NMR spectra of HA and the cross-linking product with citric acid according to Procedure I. Comparing these spectra, the appearance of an additional signal is evident in the form of multiplet with a shift of 2.24 and 2.44 ppm. CA builds into the structure of the polysaccharide and forms network of hydrogen bonds with polysaccharide chains. Literature data [58,59] have shown that CA exhibits exactly the same set of signals, with chemical shifts ranging from 2.50 ppm to 2.76 ppm. The integration of the signals on the spectrum indicates that the cross-linking of HA with CA proceeded using the hydroxyl groups from the sugar ring and the carboxyl groups of the citric acid.

The value of the coupling constant increased from 10.99 to 11.60, suggesting that a partial overlap of cross-linking agent signals took place. It is also possible that this value increased due to the formation of a network of permanent hydrogen bonds between the citric acid molecules and HA, forming physical bonds between the chains (observed as a more compact structure).

An attempt to cross-link HA with 5% CA (Figure 4) led to the obtainment of a material that was spectrally similar to the experimental material obtained with Procedure I of cross-linking HA with CA. The multiplet was observed in the range of 2.24–2.49 ppm. The remaining signals were found to overlap with peaks of unmodified HA. Regarding the integration of the signals, it is very clear that the degree of cross-linking in this case was moderate, with only one proton from citric acid for every three protons from the N-acetyl group. This was probably due to the low concentration of cross-linking agent in this procedure. 

The last attempt to cross-link HA with CA was performed in an experiment using 20% CA (Figure 5).

The obtained ^1^H-NMR spectrum was similar to that of the material obtained with Procedure I. A doublet of doublets, typical of citric acid, was visible. Other than that, no significant changes in chemical shifts were observed. In the case of signal integration, it can be seen that the value of conjugated doublets was increased, especially in comparison with the value for the sample cross-linked with the same procedure but a lower concentration of cross-linker (where the value was 1.01). In this case, there were five protons from citric acid for every three protons in the N-acetyl group. This may indicate a rather high degree of cross-linking. The consideration of the 11.76 sugar ring protons confirmed the presence of additional protons. The high concentration of CA may have been the reason for the observed results.

##### Cross-Linking with BDDE 

A final experiment on HA cross-linking was performed using BDDE as the cross-linking agent. The analysis of the ^1^H-NMR spectrum of the product of cross-linking HA with BDDE (Figure 6) showed that the signals from HA were comparable to materials obtained with CA and 1,4-diaminobutane.

The peak at 1.46 ppm indicates the presence of the CH_2_ group, which is present in the BDDE molecule [60]. A peak at 1.03 ppm seen only on the spectrum of cross-linked HA using BDDE indicates a change in the HA structure. Other characteristic signals indicating a change in HA structure under the influence of the cross-linking agent include signals around 3 ppm (3.06 and 3.30 ppm) and a higher intensity peak at 4.79 ppm [61].

#### 3.1.2. IR Spectra Analysis

All the infrared spectra of the tested materials were layered on top of the spectrum of unmodified HA. Since the spectrum of the unmodified HA was identical to the literature data [62], it was assumed that the structure and properties of the HA modified using different cross-linking agents would provide measurable results. In all the spectra shown below, the black line represents the spectrum of unmodified HA and the red line indicates the spectra for the samples after the cross-linking process. 

##### Cross-Linking with 1,4-diaminobutane

The first analyzed material was HA cross-linked with 1,4-diaminobutane in the presence of EDC and NHS, which was expected to ultimately lead to the formation of amide bonds between the carboxyl groups of HA and the amine groups of 1,4-diaminobutane. The IR spectrum (Figure 7) showed an intense characteristic signal at 1600 cm^−1^, which can be attributed to the amide group formed by the cross-linking process [63]. The peaks appeared at similar positions for both samples, indicating the effectiveness of the modification. The change in peak shape and absorbance value in the range of 3000–2500 cm^−1^ indicates a transformation of the carboxyl group (literature values of 3300–2500 cm^−1^) and a partial change in the surrounding hydroxyl groups in the polysaccharide (literature values of 3550–3200 cm^−1^) [64].

##### Cross-Linking with Citric Acid 

Citric acid, as mentioned previously, cross-links HA by forming a hydrogen bond network using the hydroxyl and carboxyl groups of CA and HA, though it is also possible to form a hydrogen bond with the N-acetyl group. Samples obtained by cross-linking HA with citric acid through two different procedures with similar cross-linking agent concentrations (25% and 20%, respectively) resulted in similar IR spectra (Figure 8). In the unmodified hyaluronic acid, there was a broad band in the range of 3500–3000 cm^−1^ characteristic of compounds containing a carboxyl group. Absorbance in this range for the cross-linked samples was lower. Such a result may indicate either the use of the carboxyl group of HA to form a bond with the cross-linker or the formation of dimers of CA through the carboxyl groups of the cross-linking agents. At 1400–1200 cm^−1^, the absorbance signal of the cross-linked samples increased. The intensified signal in this range may indicate the overlapping of hydrogen bonds formed upon effective physical cross-linking with citric acid. The 1210–1163 cm^−1^ region corresponded to C-OH stretching vibrations and is characteristic of citric acid.

Figure 9 shows the FT-IR spectra of the CA-cross-linked samples with different concentrations obtained with Procedure II. As described previously, the two most important regions are the region of about 3500–3000 cm^−1^, corresponding to the vibrations of the carboxyl and hydroxyl groups, and the region of 1400–1200 cm^−1^, resulting from the superposition of hydrogen bonds formed upon physical cross-linking with citric acid. Due to the increase in the absorbance value of the signal in the second region and the decrease in the first region, it can be concluded that the use of 5% CA made it possible to obtain cross-linked HA, though with lower efficiency. Both of these changes may also have resulted from the formation of interactions between CA molecules.

##### Cross-Linking with BDDE

The spectrum obtained for HA cross-linked with BDDE was significantly different that that obtained for pure hyaluronan (Figure 10). A characteristic band in the range of 3600–2800 cm^−1^ could be observed, and the absorbance for the cross-linked HA was much higher. These results confirmed the presence of the alkyl chain of the used cross-linker [65]. A small additional signal was also observed at 2900 cm^−1^, which corresponded to C-H stretching vibrations in the cross-linker.

Another characteristic band at 1400 cm^−1^ may have originated from a group derived from 1,4-butanediol diglycidyl ether (BDDE), as the range below 1500 cm^−1^ showed bands of stretching vibrations of C-C, C-O, and C-N single bonds and many bands corresponding to the deformational vibrations. This means that a chemical modification reaction occurred between the HA chains and BDDE molecules, resulting in the formation of a new bond network. This analysis clearly indicates the success of the cross-linking reaction via this method.

### 3.2. Transfer Optimal Method for Cross-Linking into the Procedure of the Surface Functionalization of Carbon Nonwoven Fabric

Our experiments on the cross-linking of hyaluronic acid established that the cross-linking process with 1,4-diaminobutane and EDC in the presence of NHS occurred with a relatively low efficiency (based on IR spectrum analysis). The method with BDDE proved to be uncomplicated and efficient (based on NMR and IR results). However, BDDE is a toxic compound, although it has been proven to be harmless to the human body after complete cross-linking. Finally, the reaction of the cross-linking of hyaluronan with citric acid proved to be very effective. It turned out that the higher the concentration of the cross-linking agent, the better the result. The temperature of the reaction had little effect on the process. The analysis of the final products confirmed the obtainment of cross-linked HA for all variants using CA. It can be concluded that citric acid is an eco-friendly cross-linking agent that is non-toxic, easy to use, biodegradable, biocompatible, and yields satisfying cross-linking results. CA can be successfully used in biomedical applications. Regarding the sample preparation, preliminary research showed that sublimation drying provided better results—a rigid layer of hyaluronic acid was formed on the surface of the material. Overall, the procedures with BDDE and citric acid at different concentrations (Procedure II) were chosen as the methods tested on the carbon nonwoven.

#### 3.2.1. Unmodified HA

In the initial phase of research on the formation of layers with HA coatings on the carbon nonwoven fabric, unmodified HA was used. The carbon nonwovens were treated with a 1% HA solution. Water was used as the solvent due to the polar nature of HA. However, the hydrophobic nature of the carbon nonwoven indicated that it would be better to use as lipophilic a medium as possible. The process of immersing the carbon nonwoven into the HA solution lasted 24 h. We used the standard dip-coating method. For the obtained samples, IR analysis was performed (Figure 11) to confirm the presence of hyaluronic acid on the surface of the carbon nonwoven fabric.

No characteristic bands for HA could be observed in the obtained spectrum. By skipping the cross-linking process, no hyaluronic acid layer was obtained on the surface of the carbon material.

#### 3.2.2. CA-Cross-Linked Hyaluronic Acid

In the following stages, attempts were made to develop a procedure for coating (forming a surrounding layer) the nonwoven with CA-cross-linked HA. The experiments began with the use of a low concentration of HA (0.5%), which was subsequently increased. Figure 12 shows FT-IR spectra of carbon nonwovens with a layer of CA-cross-linked hyaluronic acid.

The low concentration of hyaluronic acid and the cross-linking process with 5% CA allowed for the formation of an HA layer on the carbon nonwoven fabric. However, the absorbance values were very low, indicating the low efficiency of the process. The FT-IR spectrum showed the presence of bands characteristic of native hyaluronic acid: 3716–2993 cm^−1^ corresponding to vibrations of hydroxyl groups and secondary amide bonds (N-acetyl groups in HA), as well as a band at 1500 cm^−1^ characteristic of the C=O group of carboxylic acids and symmetric -OH groups (1402 cm^−1^) and C-O-C bonds (1021 cm^−1^). However, the use of a 5% CA solution was not sufficient. The spectrum of the sample cross-linked with citric acid at a higher concentration (20%) was clearer, with higher absorbance values. The observed characteristic bands were: Band at 1701 cm^−1^ corresponding to C=O stretching vibrations from the carboxyl group;Characteristic peaks of 1434 cm^−1^ and 1379 cm^−1^ originating from -OH and C-O-C, respectively;Peak at 1311 cm^−1^ originating from C-O bonds, confirming the cross-linking of HA;Low-intensity peaks in the 3600–3000 cm^−1^ range, indicating the use of oxygen groups during the cross-linking process.

In the next stage of research, attempts were made in order to check how the coating efficiency of carbon nonwovens would be affected by using a higher concentration of HA. Experiments were performed using 1% and 2% HA and 5% and 20% CA. The FT-IR spectra of carbon nonwovens coated with HA (coatings made using 1% HA and 5% and 20% CA are shown in the Appendix A).

The obtained spectra for samples of carbon nonwoven fabric coated with 1% hyaluronic acid and cross-linked with citric acid did not significantly differ. Only slightly higher absorbance values were observed for the samples after cross-linking with the reactant of a higher concentration. As in the previous case, characteristic bands were observed for the cross-linked hyaluronic acid and the cross-linking agent.

Figure 13 shows the FT-IR spectra of carbon nonwovens coated with HA (2%) cross-linked with CA (5% and 20%).

Interestingly, at the highest concentration of hyaluronic acid, a higher concentration of the cross-linking agent did not lead to the best results. The more intense peaks in characteristic wavelengths, as well as their shapes, on the spectrum for the sample cross-linked with 5% CA indicate that this cross-linking agent was more useful at lower concentrations. This may indicate an overloading of HA, as well as the carbon material itself, with citric acid.

#### 3.2.3. Cross-Linked HA with BDDE

In the final stage of the study of coating the carbon nonwovens with a layer of cross-linked HA, BDDE was used as the cross-linking agent. In this case, an alkaline solution of hyaluronic acid (1% NaOH solution in water) was prepared. Then, the wetted carbon material was placed in a flask for about 20 min. After this time, BDDE was added and stirring continued for another 4 h in a water bath at 45 °C. Then, the solution was diluted with distilled water, and finally PBS (pH 7.4) was added. For the purpose of analysis, a lyophilizate was obtained. Figure 14 shows the spectra of the HA-coated carbon material that was cross-linked with BDDE. It can be observed that all the obtained peaks had a low intensity. However, it is possible to identify a characteristic band in the range of 3500–3000 cm^−1^ corresponding to vibrations of acidic oxygen groups, as well as relatively intense bands of about 1260 cm^−1^ and about 1000 cm^−1^ originating from the symmetric and asymmetric vibrations, respectively, of the epoxy group found in BDDE.

Additionally, in the case of coating carbon nonwoven fabric with a layer of HA cross-linked with BDDE, different concentrations of HA (0.5%, 1% and 2%) were tested. The concentrations of BDDE per HA were: 1%, 2% and 4%, respectively. The IR spectra are shown in the Appendix A.

Figure 15 shows the IR spectra of the layers surrounding the carbon materials, which were based on HA cross-linked under different conditions with different cross-linking agents.

It can be concluded that for materials obtained with HA at both concentrations of 1% and 2%, similar results were obtained regardless of the cross-linking agent. The situation was different for the samples coated with 0.5% hyaluronic acid. In this case, the procedure using citric acid with a concentration of 20% was the most optimal; the intensity and location of peaks on the spectrum showed the most effective cross-linking. BDDE as a cross-linking agent worked with a similar efficiency. In contrast, cross-linking with 0.5% citric acid did not yield the expected results, and the bands on the spectra indicate a higher content of unchanged HA.

### 3.3. Attempts to Synthesize Covalent Conjugates of BMP Fragments with HA

Bone morphogenetic proteins (BMPs) are classified as proteins that play key roles in bone remodeling—they induce intramembranous ossification and the calcification of newly formed connective tissue [66]. BMPs are characterized by osteoinductive activity, stimulating the regeneration process of cartilage tissue [67]. The osteogenic properties of BMPs have led to intensive investigation of their use in the treatment of bone fractures [68]. Currently, recombinant human BMP-2 and BMP-7 are approved by the Food and Drug Administration for use in the treatment of open fractures of long bones, non-union fractures, or surgical spondylodesis [69].

#### 3.3.1. BMP2 (241-250) Peptide Fragment H-KRMVRISRSL-OH

The first stage of the study utilized a method with EDC and NHS that is commonly used to form conjugates of peptides/polypeptides and proteins with polysaccharides [70,71,72]. This method is efficient in the synthesis of amide bonds. In this procedure, the condensing reagent (EDC and NHS) was added to a mixture of solubilized HA and a fragment of BMP2 (241–50) peptide fragment H-KRMVRISRSL-OH. Figure 16 shows the schematic representation of the reaction.

Because of the used one-pot procedure, it is difficult to say which functional groups of HA and the peptide were involved in the synthesis of the amide bonds. The structure of the final conjugate of HA fragment BMP2 was examined with IR (Figure 17) and MALDI-TOF MS (Figure 18).

Analyzing the above spectrum, it was concluded that:The shape of the spectrum obtained for the used peptide was characterized by the highest intensities of the individual bands. A decrease in the absorbance values of the conjugate bands in the region of 1700 cm^−1^ (C=O carbonyl vibrations) and about 1300 cm^−1^ (C-O vibrations) indicated the formation of a bond with hyaluronic acid.There was a decrease in the absorbance value of the characteristic 3500–3000 cm^−1^ band derived from native hyaluronan, indicating that the carboxyl groups of the acid were used to form an amide bond with the peptide.There was an increase in the intensity of the bands at 1700 cm^−1^ when comparing pure HA to the conjugate. This was due to the incorporation into the structure of the acid, the polypeptide chain, and the formation of an amide bond between them.

Based on the above data, it can be concluded that the obtained product was a material with the expected structure. A clear effect of modification on the structure of sugar was observed. 

For the MS analysis, we used an HCCA matrix (α-cyano-4-hydroxy cinnamic acid derivative), which is adequate for samples with expected masses of up to 3500 g/mol. The analysis was based on the average molar mass of the HA monomer reacting with the maximum yield.

The molar mass calculated for the product of the chemical modification of hyaluronic acid with a peptide was 1606.77 g/mol (assuming a 100% process efficiency). An analysis of the individual peaks is included in Table 1.

Figure 19 shows the mass spectrum obtained for the native hyaluronic acid. Table 2 provides an interpretation of the most important peaks.

The theoretical molar mass of the hyaluronic acid monomer was 379.32 g/mol. The obtained spectrum confirmed the expected structure of the hyaluronic acid. Minor deviations from this value for other peaks may have been due to the presence of trace impurities of water, oxygen, sodium, or potassium (resulting from instrument specifications).

#### 3.3.2. BMP9 (361-370) Peptide Fragment H-FFPLADDVTP-OH

There were also attempts to synthesize a conjugate of HA with a fragment of the BMP9 (361-370) peptide fragment with the H-FFPLADDVTP-OH sequence. This time, 4-(4,6-dimethoxy-1,3,5-triazin-2-yl)-4-methylmorpholinium-4-toluene sulfonate (DMT/NMM/TosO^-^) was used as the condensing reagent. This reagent was developed at the Institute of Organic Chemistry, Technical University of Lodz, Poland, and it is now counted among the new generation of triazine coupling reagents [73,74,75]. It is characterized by an exceptionally high efficiency in the formation of amide bonds. Moreover, the synthesis of peptides in the presence of DMT/NMM/TosO^-^ does not require the use of additional reagents to prevent their racemization, and the condensation reagent itself and the by-products of the condensation reaction have very good solubility in most solvents used in SPPS. In addition, this reagent is also efficient in the synthesis of amides and esters in solution [76].

In this case, the synthesis strategy was changed. In the first step, the BMP9 (361-370) peptide fragment H-FFPLADDVTP-OH was treated with DMT/NMM/TosO^-^, which finally allowed for the formation of the peptide’s triazine ester. In the used peptide, there were three carboxyl groups capable of reacting with the triazine condensing reagent (in the C-terminal proline residue) and two in the side chains of the two aspartic acid residues). However, the α-carboxylic group was expected to be the most reactive of these. In addition, no excess condensing reagent was used so as not to promote the double activation reaction of the carboxyl groups. After the formation of the triazine ester of the peptide (controlled by TLC, where the disappearance of DMT/NMM/TosO^-^ with Rf = 0 stained with NBP was observed), the resulting active peptide was added to the HA solution. In hyaluronic acid, the only groups with which a condensation reaction can take place are hydroxyl groups, so the final conjugate was expected to contain a peptide linked to HA by ester bonds. Figure 20 shows the reaction scheme of the synthesis of the conjugate with the triazine reagent. 

The final product was analyzed using IR (Figure 21) and MS (Figure 22).

After analyzing the above spectrum, it was concluded that:The shape of the spectrum obtained for the pure peptide was similar to the shape of the spectrum of the polysaccharide–peptide conjugate. The reductions in the absorbance values of the bands in the regions of 1700–1600 cm^−1^ (C=O carbonyl vibrations) and about 1300 cm^−1^ (C-O vibrations) indicated the formation of an ester bond with hyaluronic acid.There was a shift in the characteristic signal of 3500–3000 cm^−1^ from native hyaluronan towards lower values (about 3100–2900 cm^−1^), which indicated the use of carboxyl groups of the acid to form an ester bond with the peptide.There was a significant increase in band intensity at 1700 cm^−1^ compared with pure HA. This was due to the facts that the polypeptide chain was incorporated into the acid structure and an amide bond was formed between them.

The MS (MALDI-TOF) analysis (Figure 22) of the resulting conjugate was also conducted. 

The molar mass calculated for the product of the chemical modification of hyaluronic acid with the H-FFPLADDVTP-OH peptide was 1482.56 g/mol (assuming 100% process efficiency). A high-intensity peak of 1107.563, as well as peaks corresponding to masses of 1194.633 and 1246.603 that can be considered peaks from the expected product, could be observed on the spectrum. The interpretation of the most important peaks is included in Table 3. 

### 3.4. Attempts to Synthesize Hybrid Materials Based on HA-Coated Carbon Nonwoven and a Biologically Active Fragment of BMP

In the first stage of the study, HA chemically modified with a peptide was used to coat the carbon nonwoven. The HA–KRMVRISRSL conjugate was used. The methodology for handling the modified hyaluronic acid was identical to that for the native hyaluronan. The wetted carbon material was introduced into the solution at the homogenization stage before the addition of the cross-linking reagent. To cross-link the HA–KRMVRISRSL conjugate deposited on the carbon nonwoven, CA and BDDE were used.

Infrared spectroscopy was used to confirm the structure of the cross-linked HA–peptide conjugate layer deposited on the carbon nonwoven fabric. Figure 23 shows a summary of the obtained results. 

In addition, we investigated whether it was possible to form a layer coating the carbon nonwoven composed of a mixture of HA and the biologically active peptide H-FFPLADDVTP-OH. This mixture was deposited on the carbon nonwoven and cross-linked with CA and BDDE.

Infrared spectroscopy was used to confirm the structure of the cross-linked layer composed of HA and a biologically active peptide. Figure 24 shows a summary of the obtained results.

All of the spectra shown in Figure 23 and Figure 24 prove the successful cross-linking of both the covalent HA–KRMVRISRSL conjugate and the mixture of HA and the H-FFPLADDVTP-OH peptide.

Between Figure 23B,C and Figure 24B,C, the most significant differences can be seen in the intensity of the bands indicating the effectiveness of functionalization with the protein fragment (3500–3000 cm^−1^ and about 1600 cm^−1^). Notably, the spectrum shown in Figure 24C stands out from the other spectra. The spectrum for the sample after physical modification indicates the better performance of this type of modification. This was the sample with the highest tested concentration of HA and a relatively low concentration of the cross-linking agent. This shape of the curve may therefore be a consequence of the concentrations of the used substances: highly concentrated hyaluronic gel has a large number of protons on its surface with which a low-concentration aqueous CA solution interacts very easily (presence of citrate ions), forming a hydrogen-bonding network. However, the nature of this cross-linking is impermanent—such a low concentration of cross-linker, relative to the concentration of sugar, would require a significant increase in exposure time to achieve complete and stable cross-linking. Therefore, it can be concluded that the tested approaches are effective, and obtaining the desired final product structure requires a chemical approach with optimized parameters.

## 4. Conclusions

The choice of cross-linking procedure primarily depends on the final application of a product. Table 4 presents suitability assessment of each of used cross-linking agents. Modulating parameters such as reactant concentration, temperature, and reaction time significantly affect the degree of cross-linking. It has been proven that a high concentration of a cross-linking agent results in the rapid cross-linking of HA. Hyaluronic acid’s unique insensitivity to ions due to its structural characteristics is the reason why achieving the complete cross-linking of the polysaccharide requires extended exposure time regardless of the type of cross-linking (physical/chemical). 

Here, the most effective HA cross-linking methods were used to coat/deposit HA or HA–peptide conjugates (BMP fragment) on the surface of carbon nonwoven fabric. However, we demonstrated that the use of covalently modified hyaluronate–peptide conjugate was significantly more efficient than the formation of a layer containing sugar and peptide components applied layer by layer. 

The obtained hybrid materials based on carbon nonwovens coated with HA or the conjugate of an HA-biologically active peptide may be applied as scaffolds for tissue regeneration, and the used approach can significantly improve the adhesion of such a material due to its high biological compatibility and ease of working in the environment of wet biological tissue. The effects of such materials on the adhesion and proliferation of osteoblasts and chondrocytes will be the subjects of further studies.

## Figures and Tables

**Figure 1 polymers-15-01551-f001:**
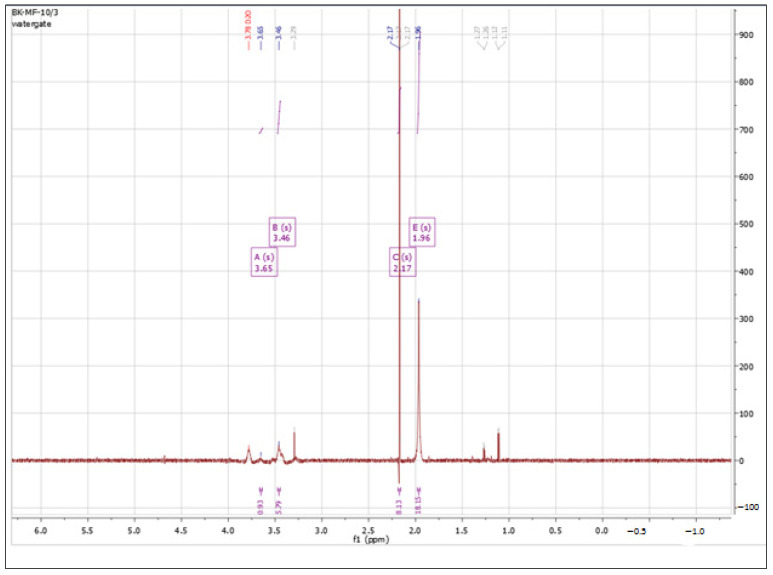
^1^H NMR spectrum of unmodified HA in D_2_O. Spectrum recorded with MestReNova 9.0 software.

**Figure 2 polymers-15-01551-f002:**
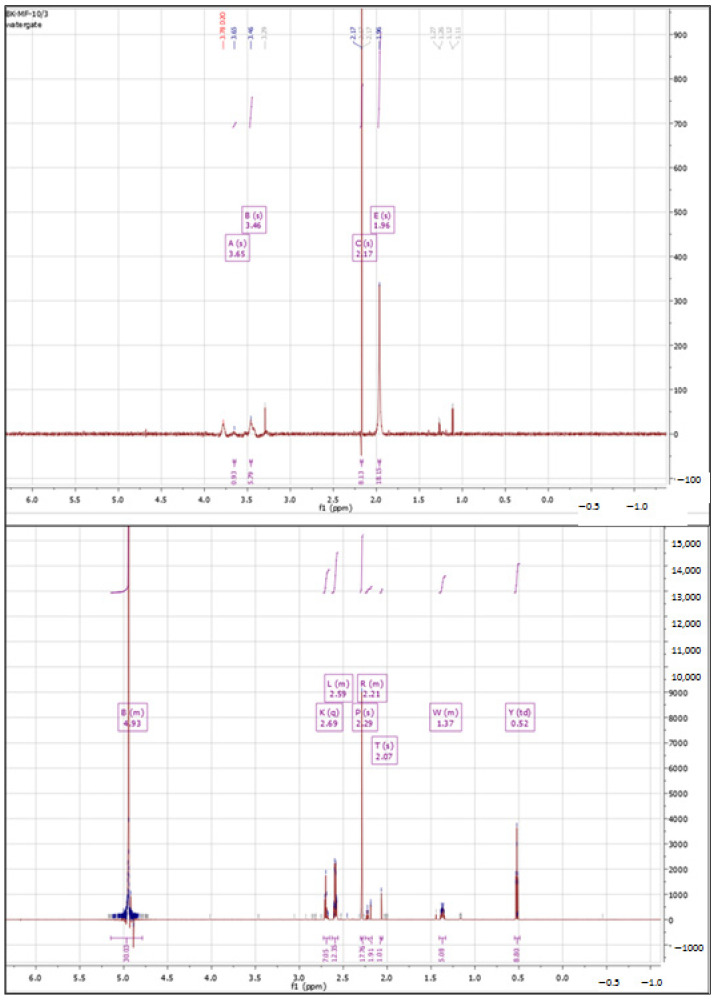
^1^H NMR spectra of native HA (**top**) vs. HA cross-linked with butanediamine (**bottom**).

**Figure 3 polymers-15-01551-f003:**
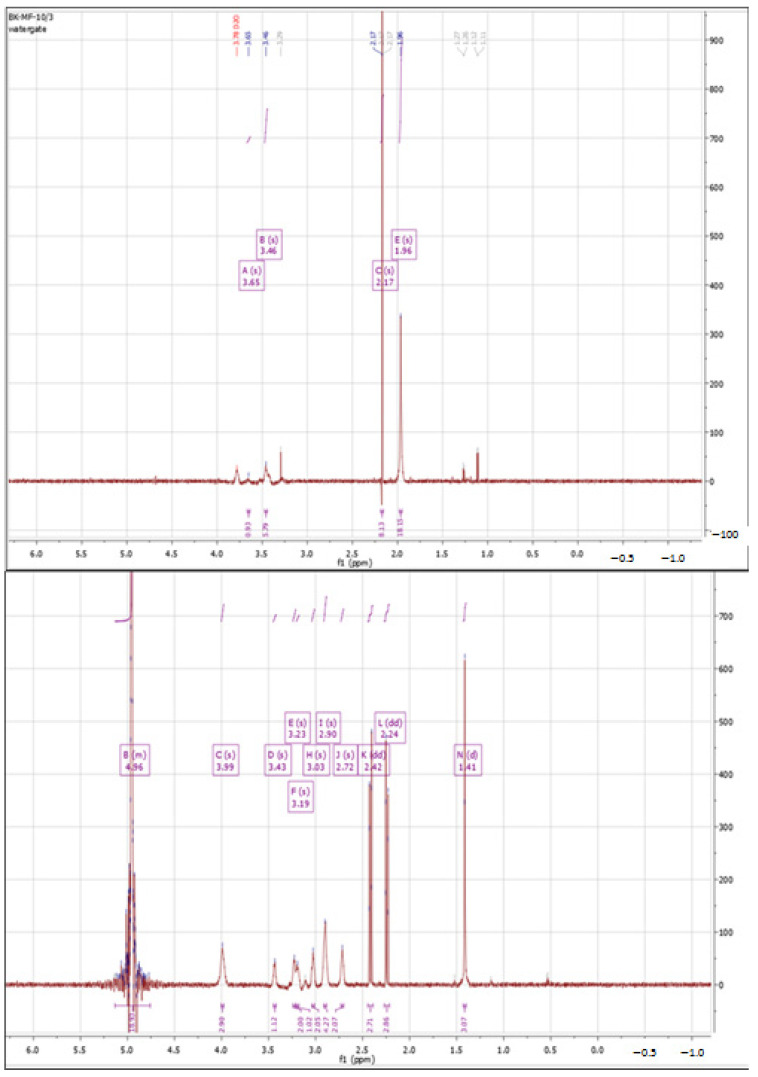
^1^H NMR spectra of native HA (**top**) vs. HA cross-linked with citric acid via Procedure I (**bottom**).

**Figure 4 polymers-15-01551-f004:**
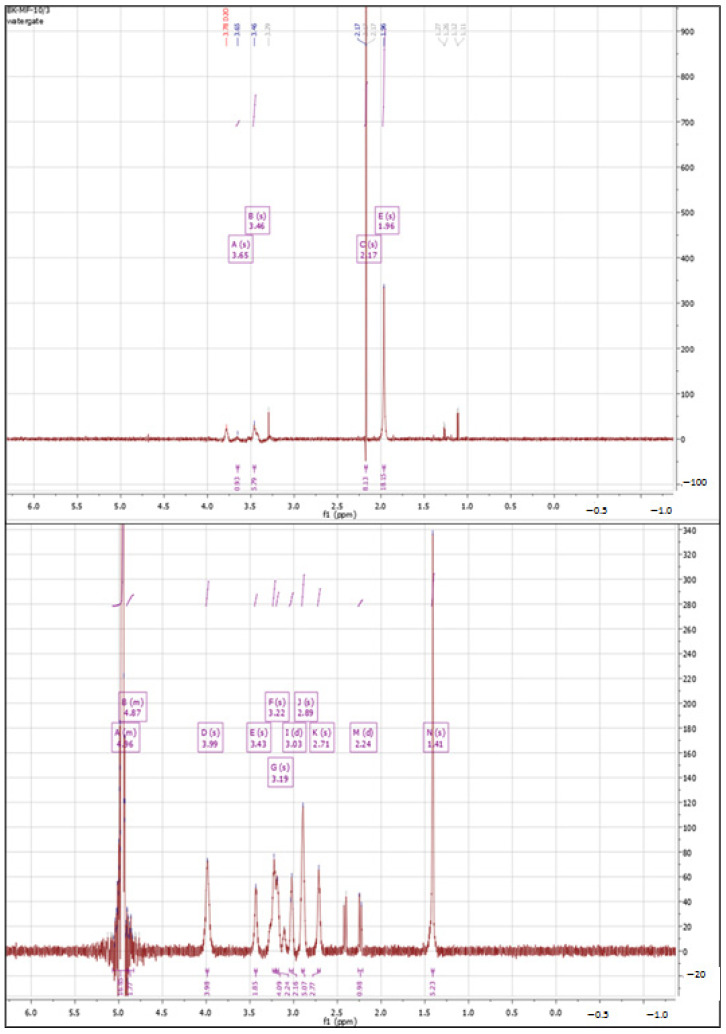
^1^H NMR spectra of native HA (**top**) vs. HA cross-linked with 5% citric acid via Procedure II (**bottom**).

**Figure 5 polymers-15-01551-f005:**
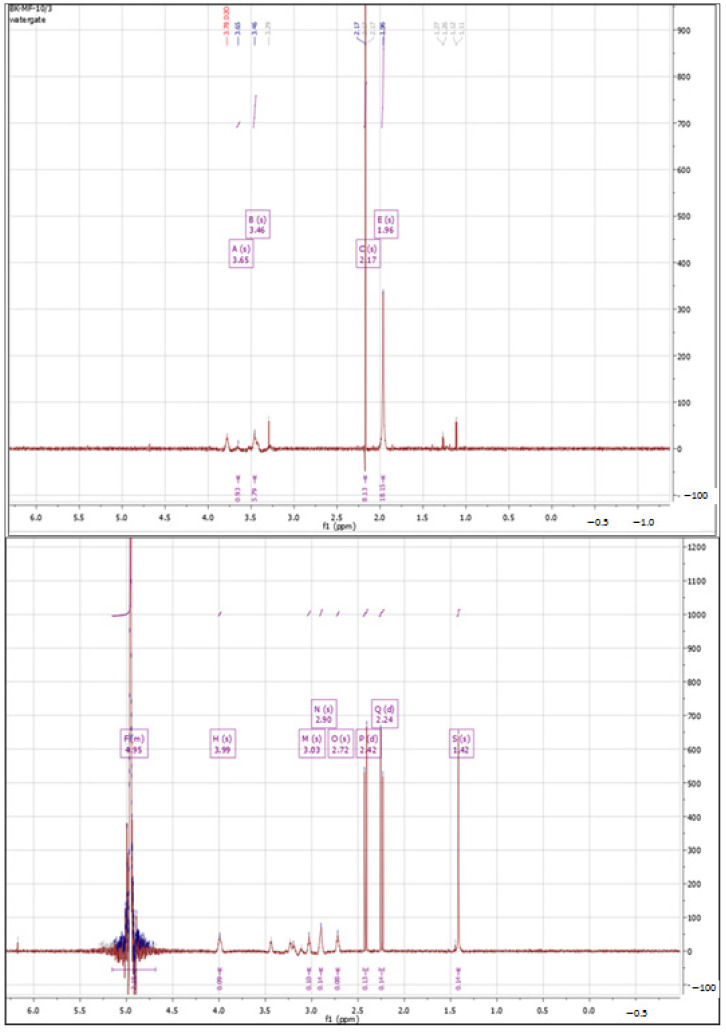
^1^H NMR spectra of native HA (**top**) vs. HA cross-linked with 20% citric acid via Procedure II (**bottom**).

**Figure 6 polymers-15-01551-f006:**
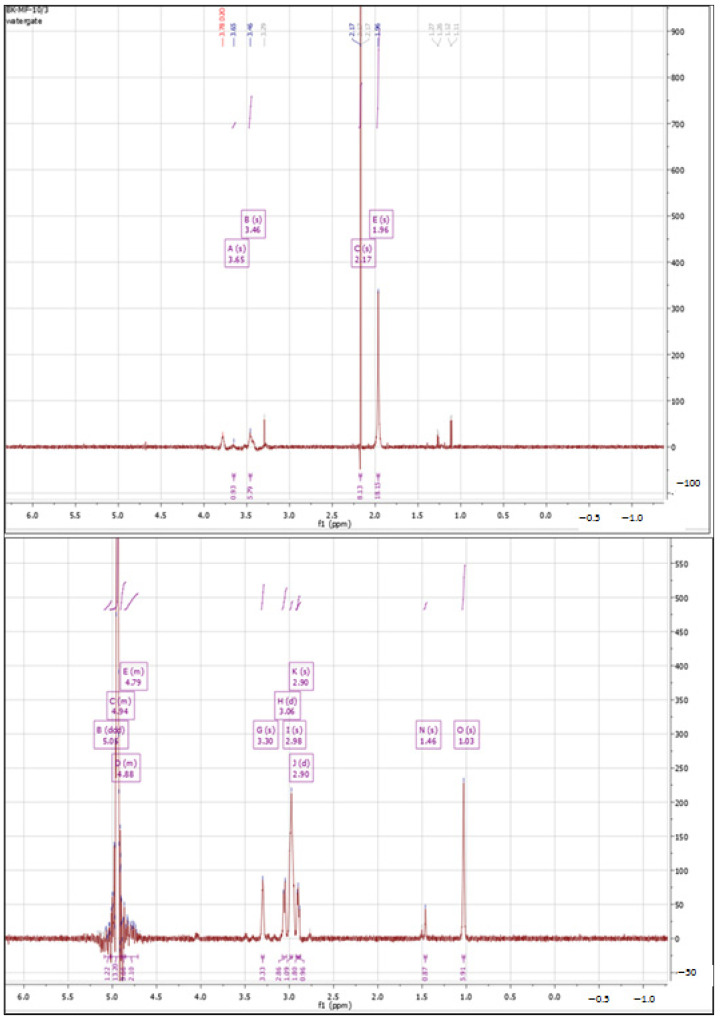
^1^H NMR spectra of native HA (**top**) vs. HA cross-linked with BDDE (**bottom**).

**Figure 7 polymers-15-01551-f007:**
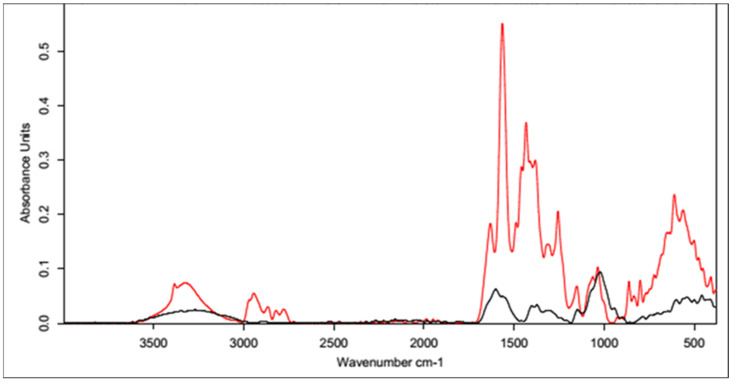
FT-IR spectrum of native HA (black) vs. HA cross-linked with 1,4-butanediamine (red).

**Figure 8 polymers-15-01551-f008:**
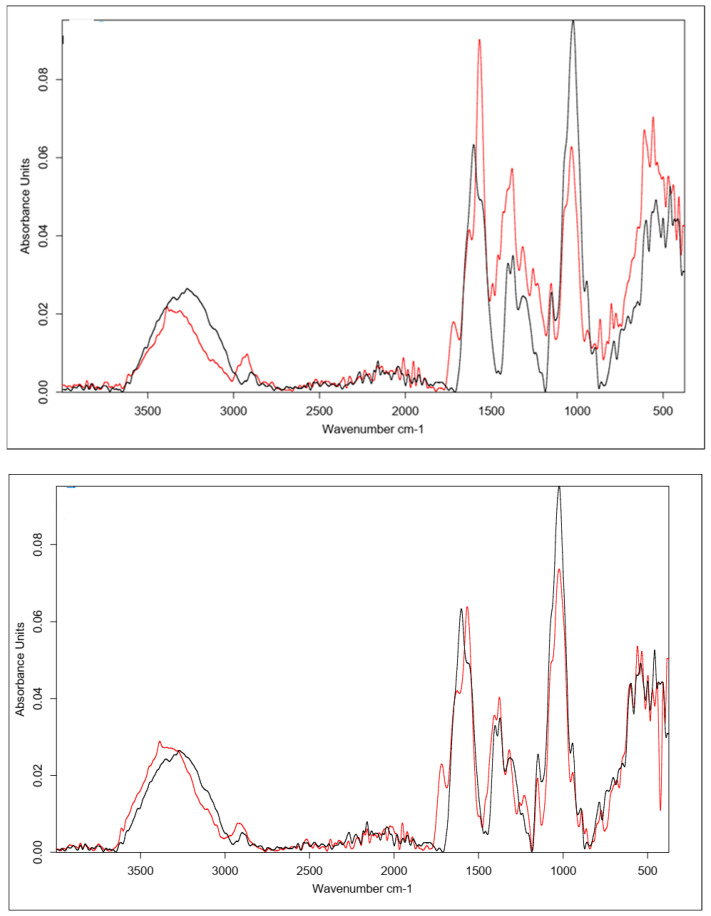
FT-IR spectra of the HA cross-linked with CA via both procedures (Procedure I—top one; Procedure II—bottom one). FT-IR spectrum of native HA (black) vs. HA cross-linked with 25% citric acid via Procedure I (top, red) and 20% citric acid via Procedure II (bottom, red).

**Figure 9 polymers-15-01551-f009:**
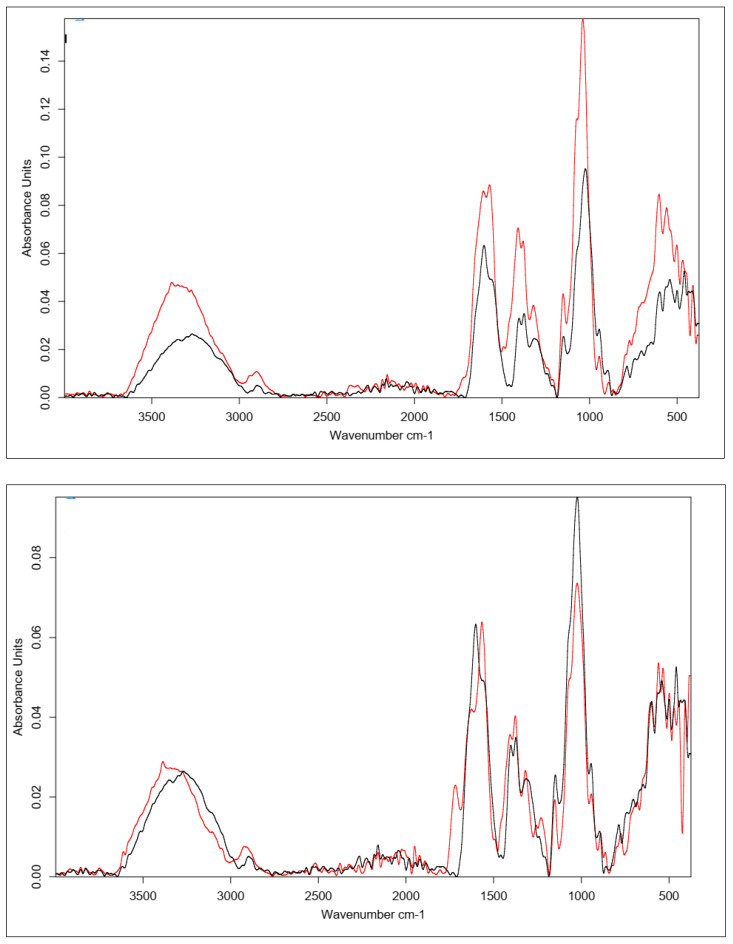
FT-IR spectra of HA cross-linked with CA via Procedure II with different concentrations of the cross-linking agent. FT-IR spectrum of native HA (black) vs. HA cross-linked with citric acid via Procedure II: 5% (top, red) and 20% (bottom, red).

**Figure 10 polymers-15-01551-f010:**
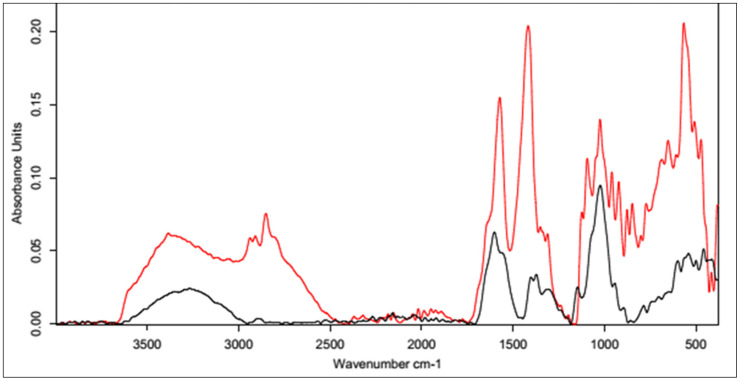
FT-IR spectrum of native HA (black) vs. HA cross-linked with 1,4-butanediol diglycidyl ether, BDDE (red).

**Figure 11 polymers-15-01551-f011:**
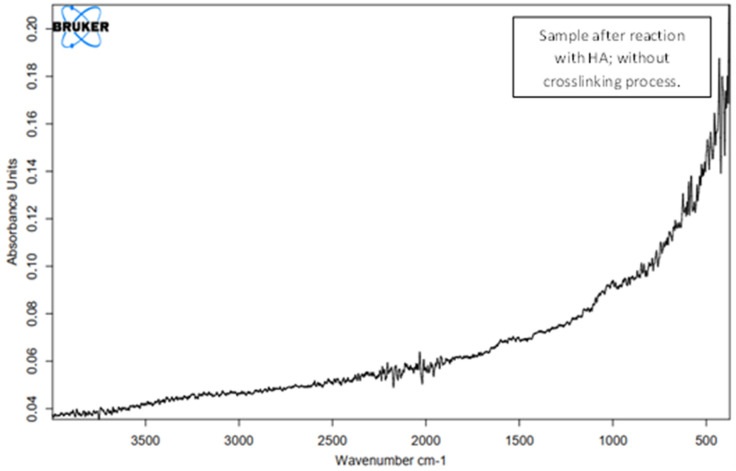
IR spectrum for a carbon nonwoven fabric after treatment with hyaluronic acid without the cross-linking process.

**Figure 12 polymers-15-01551-f012:**
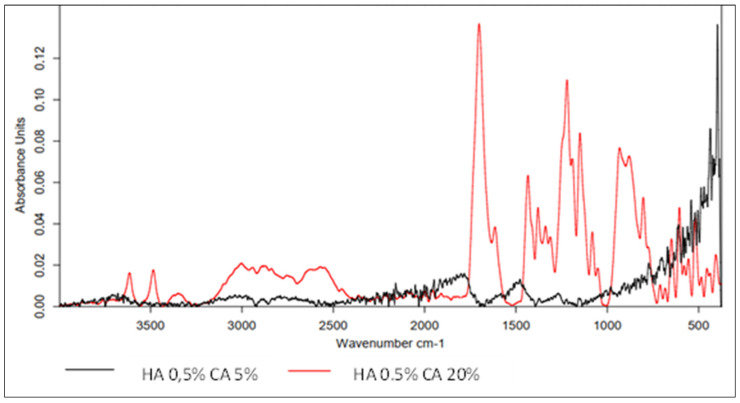
IR spectrum for 0.5% hyaluronic-acid-coated carbon nonwoven samples cross-linked with citric acid of different concentrations (5%—black line; 20%—red line).

**Figure 13 polymers-15-01551-f013:**
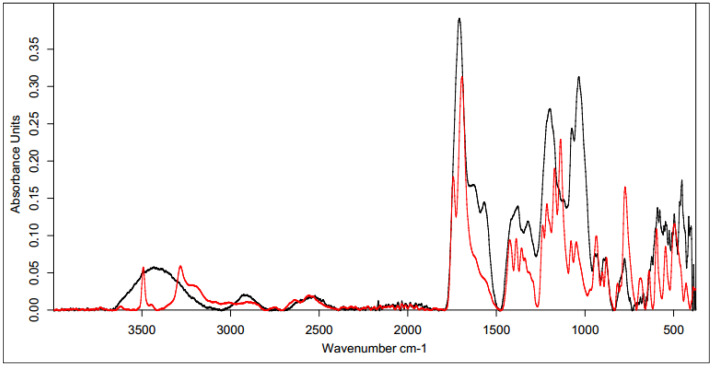
IR spectrum for 2% hyaluronic-acid-coated carbon nonwoven samples cross-linked with citric acid of different concentrations (black line: CA 5%, red line: CA 20%).

**Figure 14 polymers-15-01551-f014:**
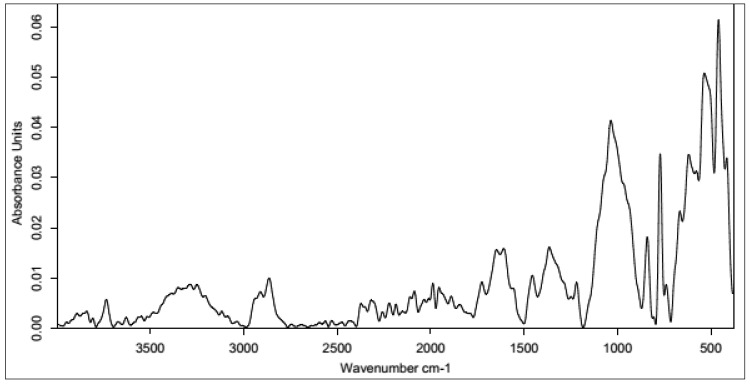
IR spectrum for hyaluronic-acid-coated carbon nonwoven samples cross-linked with BDDE.

**Figure 15 polymers-15-01551-f015:**
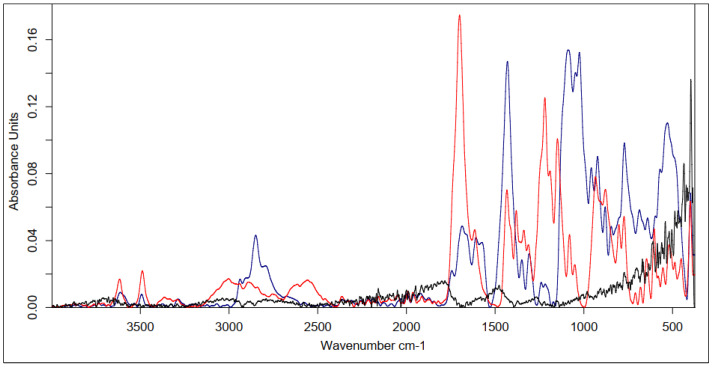
Comparison of IR spectra obtained for carbon nonwovens coated with hyaluronic acid at concentrations of 0.5% (**top**), 1% (**center**), and 2% (**bottom**) and cross-linked with different cross-linking agents (5% CA—black line; 20% CA—red line; BDDE—blue line).

**Figure 16 polymers-15-01551-f016:**
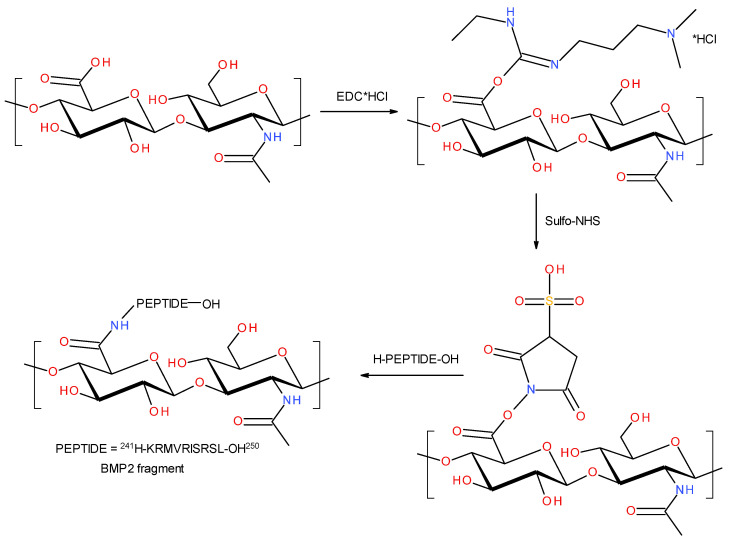
Schematic reaction of the condensation of carboxyl groups of polysaccharides with amino groups of H-KRMVRISRSL-OH by using EDC/NHS.

**Figure 17 polymers-15-01551-f017:**
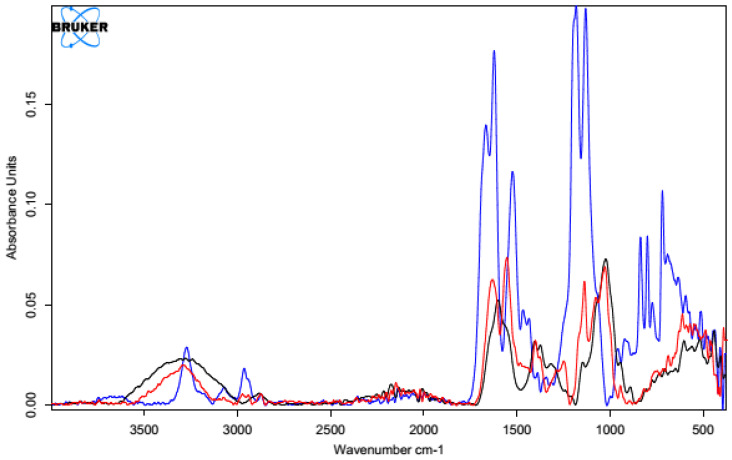
IR spectrum of native HA (black line) vs. HA–KRMVRISRSL conjugate (red line) and the spectrum of the used peptide (blue line).

**Figure 18 polymers-15-01551-f018:**
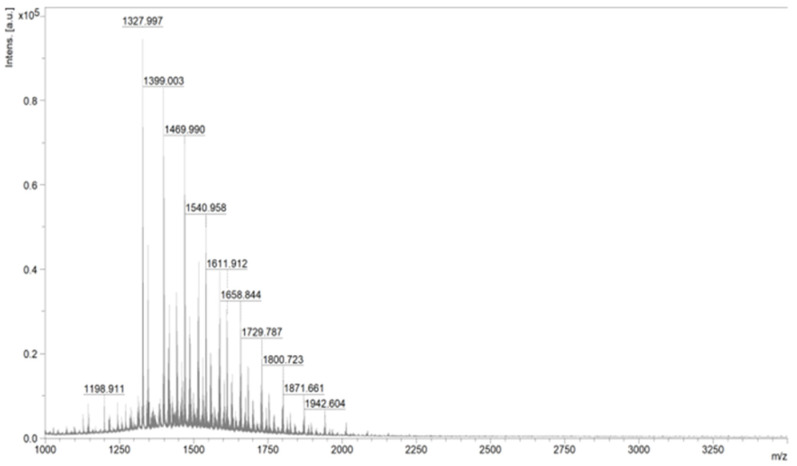
MS spectrum of the obtained polysaccharide–H-KRMVRISRSL-OH conjugate.

**Figure 19 polymers-15-01551-f019:**
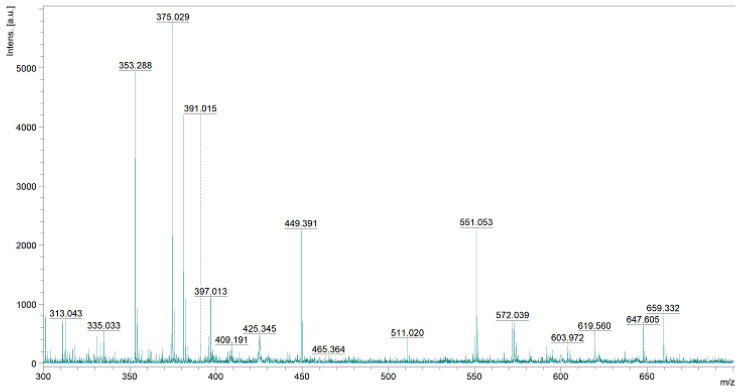
MS spectrum of the native hyaluronic acid.

**Figure 20 polymers-15-01551-f020:**
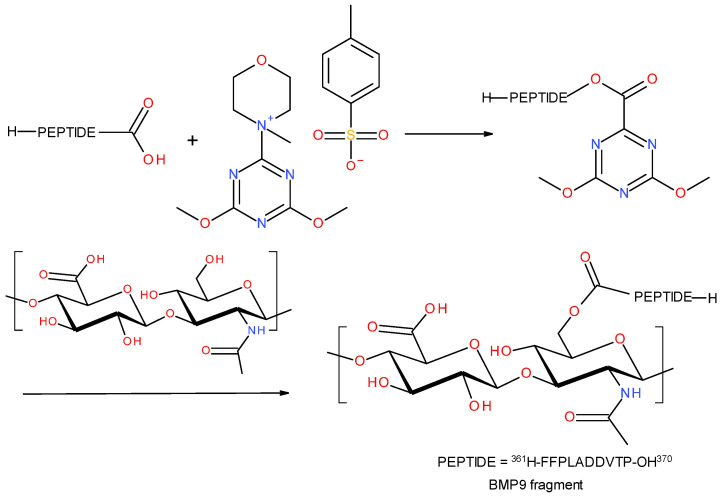
Schematic reaction of the polysaccharide–peptide condensation by using DMT/NMM/TsO^-^ as the coupling reagent.

**Figure 21 polymers-15-01551-f021:**
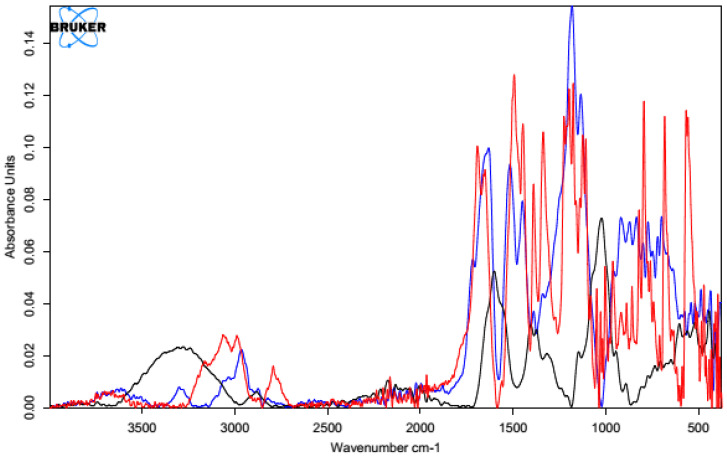
IR spectrum of native HA (black line) vs. HA–FFPLADDVTP conjugate (red line) and the spectrum of the used peptide (blue line).

**Figure 22 polymers-15-01551-f022:**
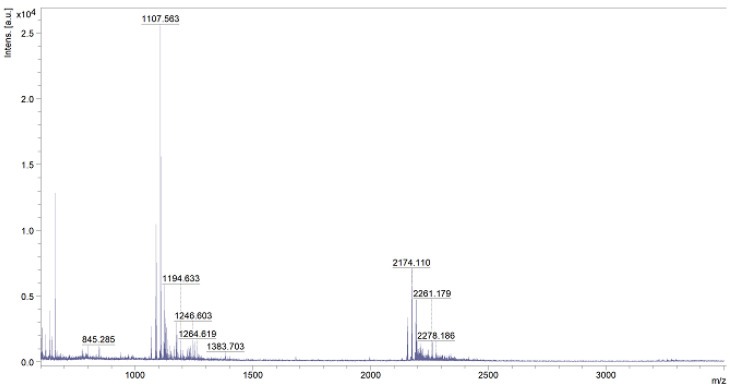
MS spectrum of the obtained polysaccharide–H-FFPLADDVTP-OH conjugate.

**Figure 23 polymers-15-01551-f023:**
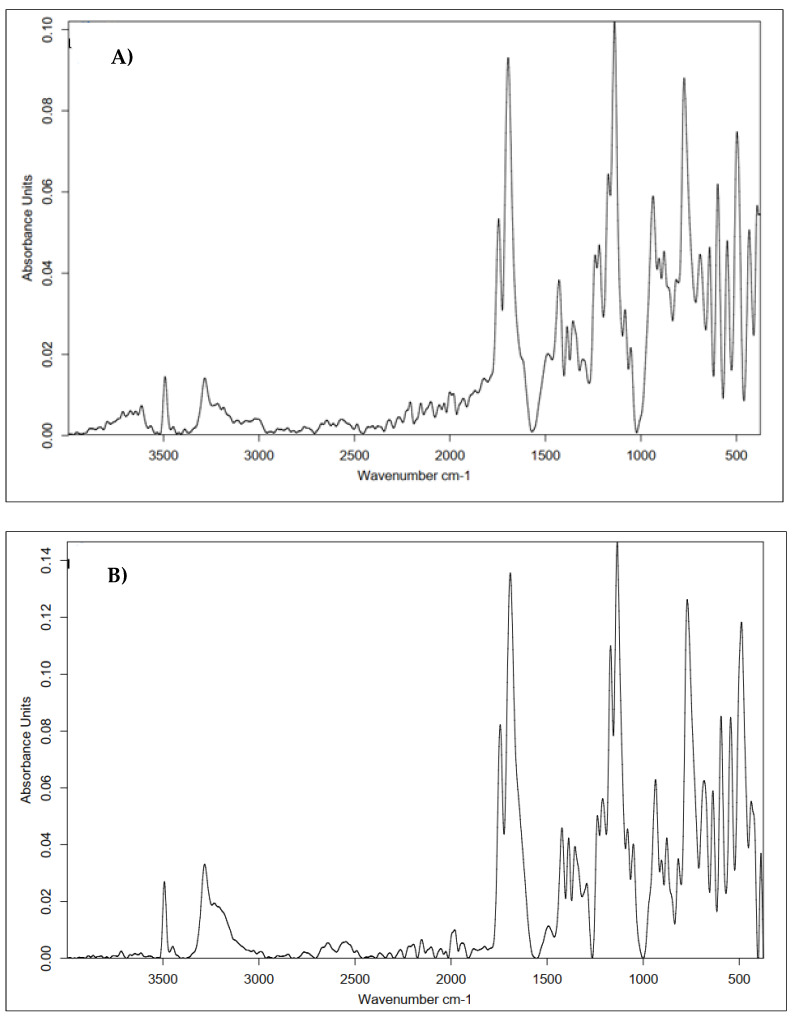
Comparison of the IR spectra obtained for the materials coated with the conjugate of hyaluronic acid–peptide (HA–KRMVRISRSL) cross-linked with different cross-linking agents. Approach: chemical modification of the coating layer. (**A**) Modified 0.5% HA and 20% CA; (**B**) modified 1% HA and 20% CA; (**C**) modified 2% HA and 5% CA; (**D**) modified 1% HA and BDDE.

**Figure 24 polymers-15-01551-f024:**
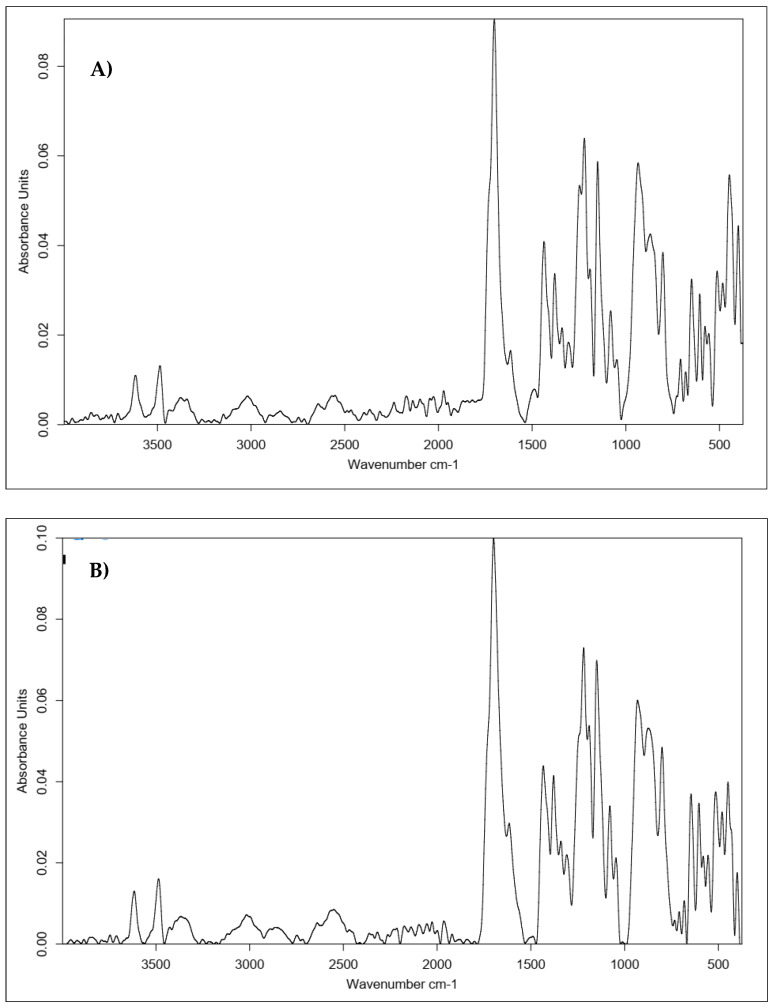
Comparison of the IR spectra obtained for the materials coated with mixture of hyaluronic acid H-FFPLADDVTP-OH and cross-linked with different cross-linking agents. Approach: layer-by-layer physical modification. (**A**) With 0.5% HA, 20 mg of peptide and 20% CA; (**B**) 1% HA, 20 mg of peptide, and 20% CA; (**C**) 2% HA, 20 mg of peptide, and 5% CA; (**D**) 1% HA, 20 mg of peptide, and BDDE.

**Table 1 polymers-15-01551-t001:** Analysis of the characteristic peaks from the MS spectrum of the obtained polysaccharide–H-KRMVRISRSL-OH conjugate.

*m*/*z*	Intensity	Interpretation
1611.912	29,253	Mass of the expected product
1658.844	27,497	
1540.958	49,859	
1198.911	9925	Mass differing by one mer
1327.997	97,107	Peptide mass
1399.003	82,979	

**Table 2 polymers-15-01551-t002:** Analysis of the characteristic peaks from the MS spectrum of native hyaluronic acid.

*m*/*z*	Intensity	Interpretation
353.288	4895	Mass of the expected product
375.029	5694	
391.015	255	

**Table 3 polymers-15-01551-t003:** Analysis of the characteristic peaks from the MS spectrum of the obtained polysaccharide–H-FFPLADDVTP-OH conjugate.

*m*/*z*	Intensity	Interpretation
1107.554	23,195	Mass of the expected product differing by one mer
1246.593	1261	
1264.605	920	
1194.623	1562	Peptide mass

**Table 4 polymers-15-01551-t004:** Summary of the results of each cross-linking agent used for the cross-linking process of HA.

Cross-Linking Agent	Suitability Assessment/Explanations
1,4-butanediamine	Mediocre results. Amine was attached to the polysaccharide chain, but results were below expectations—only a slight modification took place.
Citric Acid	Reagent with a simple structure that is able to cross-link HA either chemically or physically with good results. Considered a “green” cross-linking agent. It is non-toxic, biodegradable, and biocompatible.
BDDE	Simple method that effectively cross-links HA at a sufficient level. Judging by the obtained structure of the final material, it provides superior mechanical properties relative to CA-using methods, so the acid clumps into larger agglomerates and has a longer duration. If completely cross-linked, it is considered harmless to the human body.

## Data Availability

Not applicable.

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
