# Peer review of "Coating Methods of Carbon Nonwovens with Cross-Linked Hyaluronic Acid and Its Conjugates with BMP Fragments"

_polymers, 2023, doi:10.3390/polym15061551_

Round 1

Reviewer 1 Report

After careful reading the manuscript titled with "Coating methods of carbon nonwovens with cross-linked hyaluronic acid and its conjugates with BMP-fragments", here is a few comments which makes them more interesting to the readers. 

Need more detailed literature, which says the gaps and the objectives of this work. 

Provide the abbrevation for the BMP proteins in abstract sections. 

Provide more data on the NMR measurement (i.e., how many scans, conditions) and FTIR for scans. 

Figure 1, specify the software that used for interpreting NMR. 

Author Response

Dear Reviewer, 

I have corrected the manuscript according to your suggestions. Please see the attachment.

Kind regards, 

Sylwia Magdziarz

Reviewer 2 Report

Dear Authors

Many thanks for your interesting study about cross-linking mechanisms on hyaluronic acid. The work is complete, but I have some questions and suggestions to be applied to it.

Some mistakes in spelling or missing the whole name of the compounds are stated and marked in yellow inside the pdf. Please, try to correct or incorporated in every case.

Line 180. Please, specify the temperature of the reaction. Heated is not enough, from my point of view

From the line322, there are different formats in line space used. Please, consider to unify

Line 371. Please, try to incorporate more explanation about what AB system is.

Line 428. Fig 8 , as well as in Fig 9, there are some deviations on the signals on FTIR (in -OH range and in 1500-1700 cm-1) that should be explained in detail. In my opinion, this deviations show molecular interactions in the cross-linked substrate.

Line 746. Maybe it would be more clear for readers if you can summarize the results of each procedure in form of table and, please, extend the explanations

Conclusions??? You have not included

Many thanks

Many thanks

Author Response

(The authors gave the same response as above.)

Author Response

(The authors gave the same response as above.)

Round 2

Reviewer 3 Report

The authors have modified the manuscript according to the requirements, so I agree with its publication in the modified form.